# Effect of Germination on the Physicochemical Properties, Functional Groups, Content of Bioactive Compounds, and Antioxidant Capacity of Different Varieties of Quinoa (*Chenopodium quinoa* Willd.) Grown in the High Andean Zone of Peru

**DOI:** 10.3390/foods13030417

**Published:** 2024-01-27

**Authors:** Betsy S. Ramos-Pacheco, David Choque-Quispe, Carlos A. Ligarda-Samanez, Aydeé M. Solano-Reynoso, Henry Palomino-Rincón, Yudith Choque-Quispe, Diego E. Peralta-Guevara, Elibet Moscoso-Moscoso, Ángel S. Aiquipa-Pillaca

**Affiliations:** 1Agroindustrial Engineering, Universidad Nacional José María Arguedas, Andahuaylas 03701, Peru; dchoque@unajma.edu.pe (D.C.-Q.); caligarda@unajma.edu.pe (C.A.L.-S.); hpalomino@unajma.edu.pe (H.P.-R.); deperalta@unajma.edu.pe (D.E.P.-G.); aiquipapillaca@gmail.com (Á.S.A.-P.); 2Food Science and Technology, Universidad Nacional de San Antonio Abad del Cusco, Cusco 08000, Peru; 3Food Nanotechnology Research Laboratory, Universidad Nacional José María Arguedas, Andahuaylas 03701, Peru; elibetmm22@gmail.com; 4Nutraceuticals and Biomaterials Research Group, Universidad Nacional José María Arguedas, Andahuaylas 03701, Peru; amsolano@unajma.edu.pe (A.M.S.-R.); ychoque@unajma.edu.pe (Y.C.-Q.); 5Water and Food Treatment Materials Research Laboratory, Universidad Nacional José María Arguedas, Andahuaylas 03701, Peru; 6Research Group in the Development of Advanced Materials for Water and Food Treatment, Universidad Nacional José María Arguedas, Andahuaylas 03701, Peru; 7Department of Basic Sciences, Universidad Nacional José María Arguedas, Andahuaylas 03701, Peru; 8Department of Environmental Engineering, Universidad Nacional José María Arguedas, Andahuaylas 03701, Peru

**Keywords:** quinoa, germination, antioxidant capacity, phenols, flavonoids, functional ingredient

## Abstract

Germination is an effective strategy to improve the nutritional and functional quality of Andean grains such as quinoa (*Chenopodium quinoa* Willd.); it helps reduce anti-nutritional components and enhance the digestibility and sensory aspects of the germinated. This work aimed to evaluate the effect of germination (0, 24, 48, and 72 h) on the physicochemical properties, content of bioactive compounds, and antioxidant capacity of three varieties of quinoa: white, red, and black high Andean from Peru. Color, nutritional composition, mineral content, phenolic compounds, flavonoids, and antioxidant activity were analyzed. Additionally, infrared spectra were obtained to elucidate structural changes during germination. The results showed color variations and significant increases (*p* < 0.05) in proteins, fiber, minerals, phenolic compounds, flavonoids, and antioxidant capacity after 72 h of germination, attributed to the activation of enzymatic pathways. In contrast, the infrared spectra showed a decrease in the intensity of functional groups –CH–, –CH_2_–, C–OH, –OH, and C–N. Correlation analysis showed that flavonoids mainly contributed to antioxidant activity (r = 0.612). Germination represents a promising alternative to develop functional ingredients from germinated quinoa flour with improved nutritional and functional attributes.

## 1. Introduction

Quinoa (*Chenopodium quinoa* Willd.), an Andean cereal, has positioned itself as a superfood of increasing popularity globally because of its excellent nutritional quality due to its balanced nutrient composition [1]; it contains significant amounts of protein (13.8 to 16.5%) of high biological value [2], with all essential amino acids such as phenylalanine (0.1–2.7 g/100 g), histidine (1.4–5.4 g/100 g), isoleucine (0.8–7.4 g/100 g), threonine (2.1–8.9 g/100 g), leucine (2.3–9.4 g/100 g), lysine (2.4–7.5 g/100 g), methionine (0.3–9.1 g/100 g), tryptophan (0.6–1.9 g/100 g) and valine (0.8–6.1 g/100 g) in ideal proportions for human nutrition [3,4]. Likewise, it is characterized by its high content of micronutrients such as iron, zinc, phosphorus, magnesium, manganese, B complex vitamins, vitamin E, and phenolic compounds [5,6,7,8,9,10].

Studies have reported important health benefits associated with quinoa consumption, including anti-inflammatory, antioxidant, anticancer, antihypertensive, hypo-cholesterolemic, prebiotic, and antidiabetic properties [11,12,13,14,15,16,17]. These benefits are attributed to bioactive compounds such as polyphenols, flavonoids, phytosterols, essential fatty acids, and dietary fiber [18,19,20,21]. In addition to its benefits as a food source, quinoa has significant potential as a functional ingredient in various products. However, some anti-nutritional factors in quinoa, such as saponins and trypsin inhibitors, could limit these applications [4,22]. An effective strategy to enhance the functional attributes of quinoa is the germination of the seeds.

Controlled germination is a biotechnological process that improves the nutritional value and quality of seeds and grains. During germination, various metabolic pathways are activated that mobilize reserve nutrients, synthesize new bioactive compounds, and reduce anti-nutritional factors [23,24,25,26,27]. Specifically in quinoa, germination increases the levels of phenolic compounds, unsaturated fatty acids, γ-aminobutyrate, carotenoids, and folates while decreasing the activity of trypsin inhibitors and the concentration of phytates [28,29,30]. Thus, germination enhances the quality and health benefits associated with quinoa consumption.

Although quinoa germination has been studied as a strategy to improve its nutritional and functional quality, there are still few works focused on high Andean varieties, and they still need to carry out a comprehensive characterization of the flour properties. This knowledge is critical to determining flour’s stability, texture, and suitability in food [1,23,24].

The main objective of the present study is to evaluate the effect of germination (0, 24, 48, and 72 h) on the physicochemical properties, content of bioactive compounds, and antioxidant capacity of three varieties of quinoa white, red, and black, grown in the high Andean area of Andahuaylas, Peru. Using a comprehensive approach, this research seeks to fill the knowledge gap to develop functional quinoa ingredients with improved quality attributes. The complete experimental flow chart is shown in Figure 1.

## 2. Materials and Methods

### 2.1. Materials

The varieties of quinoa (*Chenopodium quinoa* Willd.) selected for this study (Figure 1) were Collana (black variety), Pasankalla (red variety), and an ecotype called Choclito (white variety). Collana is an original variety from the Altiplano, with small black grains, high saponin content, and a maturity cycle of 140 days. Pasankalla is a variety improved by the Instituto Nacional de Innovación Agraria—INIA, with large grains of intense red color, moderate saponin content, and a 144-day cycle. The Choclito ecotype is a traditional local variety from the department of Puno in Peru, white in color, medium grain, low saponin content, and a maturity cycle of 132 days [31]. The differences in days of maturity, stem shape, distinctive colors, and saponin content are typical attributes reported for each variety cultivated in the Andes. The samples were obtained from crops in the community of Chulcuisa, district of San Jerónimo, Province of Andahuaylas, at 3758 m of altitude; the seeds were cleaned and stored in polyethylene bags.

### 2.2. Germination

The germination process of quinoa grains was carried out following the methodology of Xing et al. [26] with some modifications. The quinoa grains were washed with plenty of distilled water and disinfected with a 1% sodium hypochlorite solution for 5 min. They were soaked in distilled water for 4 h at 20 °C until reaching a moisture content between 40 and 50% before germination.

The grains were placed in a humid chamber FOC 200 E (Velp Scientifica^TM^, Usmate Velate, Italy) at 25 °C and 95% relative humidity. After germination, the grains were collected at 24, 48, and 72 h, dried in a forced convection oven FED 115 (BINDER, Tuttlingen, Germany) at 40 °C to a moisture content below 10%, and finally ground at 150 rpm for 3 min in a cyclone mill Twister (Retsch, Haan, Alemania) and sieved into a 250 µm mesh, the germinated flour was stored in airtight glass containers for subsequent analysis.

### 2.3. Color

The color was measured using a CR-5 digital colorimeter (Konica Minolta, Tokyo, Japan); the results were obtained in CIE coordinates *L** *a** *b**, using the standard illuminant D65 and a standard observer from 10°. *L** indicates lightness from black (0) to white (100), *a** refers to shades from green (−) to red (+) and *b** from blue (−) to yellow (+). Readings were taken in reflectance modulus. The color difference (Δ*E*) between the germinated quinoa flour and the control (ungerminated) flour was calculated using Equation (1), while the whiteness index (*WI*) was calculated using Equation (2) [32,33].
(1)$$∆E={\left[{\left({∆L}^{*}\right)}^{2}+{\left({∆a}^{*}\right)}^{2}+{\left({∆b}^{*}\right)}^{2}\right]}^{1/2}$$
(2)$$WI=100-\sqrt{{\left(100-L^{*}\right)}^{2}+{a^{*}}^{2}+{b^{*}}^{2}}$$

### 2.4. Water Activity

Water activity was determined with a HygroPalm 23-AW meter (Rotronic brand, Bassersdorf, Switzerland). The instrument was calibrated with calibration standards, then 3 g of sample was taken and transferred to a disposable sample container, inserted into the chamber, and sample readings were taken [34,35].

### 2.5. Proximal Analysis

Proximate analyses of ungerminated and germinated quinoa samples were determined according to the standard methods of the Association of Official Analytical Chemists (A.O.A.C.) for moisture; the sample was dried to constant weight in a forced convection oven (AOAC 925.10). The protein was obtained using the Kjeldahl method through the destruction of organic matter with concentrated sulfuric acid (AOAC 955.04), the fat was extracted using petroleum ether in a Soxhlet extractor (AOAC 2003.05), the ash was determined by ignition at 600 °C of the sample in a muffle furnace (AOAC 942.05), and the fiber was determined by ignition of the dry residue after digestion of the sample with sulfuric acid and sodium hydroxide (AOAC 962.09). Carbohydrates were determined by difference according to Equation (3) [36].
(3)Carbohydrates=100−(moisture+protein+fat+ash+fiber)

### 2.6. Mineral Micronutrients

The content of mineral micronutrients was determined using an atomic absorption spectrometer model A6800 (Shimadzu, Kyoto, Japan). For this, 5 g of the sample was converted to ash by incineration at 550 °C for five hours. Subsequently, 0.1 g of each sample was digested with nitric acid in a microwave (SCP Science, Miniwave, Montreal, QC, Canada) oven. Finally, the elements were identified by their characteristic emission spectra compared to a standard curve [1].

### 2.7. Functional Groups

The functional groups were determined by Fourier transform infrared spectroscopy (FTIR). The Nicolet IS50 FTIR transmission module (ThermoFisher, Waltham, MA, USA) was used within a range of wave numbers between 4000 and 400 cm^−1^, 10 mg of sample was weighed, and 100 mg of potassium bromide to form the pellet in a press, the readings were taken with 32 scans and 8 cm^−1^ resolution [1,37].

### 2.8. Bioactive Compounds

The samples’ total phenolic compounds were estimated using Folin–Ciocalteu reagent. Extracts of germinated and ungerminated flour samples were prepared and mixed with 20% sodium carbonate, 0.25 N Folin–Ciocalteau reagent and deionized water. The samples were read at 755 nm on a spectrophotometer (Genesys 150, Thermo Fisher Scientific, Waltham, MA, USA). Gallic acid (GA) was used as a reference standard, and the results were expressed as mg equivalent of gallic acid/100 g of sample [38,39].

The total flavonoid content was determined using the method described by Suárez-Estrella et al. [6]. Extracts of germinated and ungerminated flour samples were mixed with methanol and aluminum chloride. The samples were read at 450 nm on a spectrophotometer (Genesys 150, Thermo Fisher Scientific, Waltham, MA, USA). The reference standard was Quercetin, and the results were expressed as mg of Quercetin equivalent/100 g of sample [40].

### 2.9. Antioxidant Capacity

The DPPH assay was performed using the stable radical 2,2-diphenyl-1-picrylhydrazyl. The samples were read at 515 nm on a spectrophotometer (Genesys 150, Thermo Fisher Scientific, Waltham, MA, USA). The sample extracts were mixed with the diluted DPPH solution, and the samples were read at 515 nm in a spectrophotometer. Trolox was used as a reference standard, and the results were expressed as mg Trolox equivalents/g sample [30,39,41,42].

### 2.10. Statistic Analysis

The results were analyzed through ANOVA, Tukey’s multiple comparison test, and Pearson correlation at 5% significance. Origin Pro 2023 software (Origin Lab Corporation, Northampton, MA, USA) was used for graphical representation and statistical tests.

## 3. Results and Discussion

### 3.1. Color

The color evaluation is shown in Table 1; germination decreased the brightness (*L**) of the germinated quinoas, while redness (*a**) and yellowness (*b**) increased with germination time. These variations would be related to the hydrolysis of starch and proteins during germination [43]. Positive values of chroma *a** and *b** are associated with carotenoid pigments in foods [44]. In this study, the germination process significantly increased both chroma parameters, more evident at 72 h. These results suggest an accumulation of carotenoids, bioactive compounds essential for their provitamin and antioxidant activity [45,46]. Specifically, Darwish et al. [47] reported increases of 26.02% in total carotenes during quinoa germination, an effect that agrees with the trend observed in the present study. Taken together, the colorimetric analysis indicates that prolonged germination times could maximize the content of these nutraceuticals in quinoa with consequent nutritional and health benefits.

The color difference (Table 1) concerning the ungerminated control exceeded the visible threshold of three units (Δ*E* > 3) [48] in all varieties and times. Increasing the germination time to 72 h generated the greatest quantifiable color changes, with values of 6.61, 5.76, and 7.65 in white, red, and black quinoa, respectively. These results show that extending germination produces measurable color variations. Furthermore, the whiteness index progressively decreased as germination progressed. The low values suggest that the flour acquired a yellowish hue, possibly due to the formation of carotenoids. White quinoa, without colored pigments, easily reflects any change in color variation and whiteness index. In contrast, black and red quinoa have base pigments that partially mask the modifications. Antioxidant substances in red and black pigments could protect other compounds from oxidative degradation during germination, causing minor color alterations [49,50,51]. Likewise, the different content of phenolic compounds in each variety influences its susceptibility to changes during germination [52].

### 3.2. Water Activity

Water activity, a parameter that quantifies water availability in food [28,40], presented its maximum values in white and red quinoa germinated for 72 and 48 h, respectively (Table 2). Although the levels obtained (water activity < 0.3) indicate that these germinated quinoa flours could be considered safe and stable ingredients for various food applications, some samples showed water activity lower than 0.2. According to Fontana [53], such low values of aqueous activity would be associated with a higher rate of lipid oxidation, a situation that could compromise the product’s useful life. Therefore, adequate control of drying kinetics is critical to obtaining a germinated quinoa ingredient with optimal levels of water availability.

### 3.3. Proximal Analysis

The results of the proximal analysis (Table 3) showed a decrease in the carbohydrate content during the germination of the quinoa varieties, an effect attributed to hydrolysis processes mediated by endogenous amylolytic enzymes, which degrade starch, stored as amylose and amylopectin into sugars simple, that is, the reducing sugars glucose and maltose and, to a lesser extent, the non-reducing sugar sucrose [24,54]. On the contrary, germination markedly increased protein levels, especially at 72 h, reaching up to 10.65% for WQ, 11.35% for RQ, and 12.98% for BQ compared to non-germinated samples. During grain germination, storage proteins are hydrolyzed into peptides and amino acids by proteolytic enzymes after 2 to 3 days from imbibition, increasing nutrient bioavailability [23,54,55]. Likewise, the results showed increases in the total fiber content after the germination process for the three varieties. Previous studies attribute these changes to the fractionation of protein structures and greater solubilization of macromolecules during the biochemical process associated with seed germination [23]. Considering the current interest in plant-based ingredients rich in dietary fiber, germination would represent an effective strategy to enhance this attribute in quinoa for various food applications [38]. Other studies have reported decreases in fat content during quinoa germination, which are found in whole grains as triacylglycerols (TGA); their mobilization requires a coordinated metabolic activity that begins with germination, leading to the net conversion of fat into sugars [24,54,56]. Although in this work, the changes in fat were minimal, it is suggested that its variation be monitored due to the impact on the stability and shelf life of the ingredient. The ash content showed a slight but significant increase during the germination process of the three quinoa varieties analyzed. The germinated BQ variety presented the highest average percentage of ashes (2.58%) after 72 h. This effect could be due to a higher relative concentration of minerals remaining as inorganic residue after the ignition of the organic matter. Regarding moisture content, although the germinated RQ presented the highest average, the kinetics of variation between times and varieties did not show a uniform trend during germination. Previous studies attribute these differences in the final moisture percentage to factors such as the water absorption rate, respiratory activity, and efficiency of the drying process for each quinoa genotype [56].

### 3.4. Mineral Micronutrients

The results showed significant increases (*p*-value < 0.05) in the content of several essential minerals, including calcium, phosphorus, iron, magnesium, and potassium, at different germination times for the three quinoa varieties studied (Table 4). These changes are attributed to the biochemical reactions and enzymatic activations typical of germination [47,57]. Furthermore, the concentration of the anti-nutrient phytate, which stores phosphorus in mature cereals, decreases during germination due to the action of phytases, thus increasing the bioavailability of phosphorus and other minerals [28,54,58]. The increases were greater than 72 h, agreeing with studies that report positive effects of prolonged germination times on the nutritional quality of quinoa. Controlled germination can increase the mineral content and nutritional functionality of different quinoa varieties.

### 3.5. Functional Groups

The infrared spectra of the germinated quinoa varieties are shown in Figure 2. A vibrational analysis was carried out through a transmission module to study the functional groups of the samples. The spectra were similar between varieties and germination times, with characteristic groups composed of –CH–, –CH_2_–, C−OH, and –OH found at wave numbers 2927, 2851, 1023, and 852 cm^−1^, respectively. Likewise, the C=O group was located at 1741 cm^−1^, possibly due to carboxylic acids or fats within the flour. The 1649 and 1543 cm^−1^ bands are attributed to amide I and II, respectively. These bands are associated with vibration modes of the amino groups of amino acids in protein structures. Therefore, they reflect modifications in the secondary protein structure [1]. In addition, the samples showed a signal at 1239 cm^−1^ (C–N vibrational mode) coming from amino acids. Analogous results were previously reported by García-Salcedo et al. [59] in quinoa, chia, and kiwicha flour and by Contreras-Jiménez et al. [1] in quinoa flour. Through analysis of the spectrograms, differences in intensities can be seen throughout the spectrum of the samples, especially between 500 and 2000 cm^−1^. The native sample exhibits higher intensities, with attenuation observed as germination time increases. This indicates increasing degradation with longer treatment time. However, no new functional group formation was evident from the germination process.

### 3.6. Bioactive Compounds

Figure 3 shows the results of total phenolics and flavonoids. The QR (44.07 mg AGE/100 g) and QN (46.96 mg AGE/100 g) presented the highest phenolic contents, while the QB (123.82 mg QE/100 g) had high levels of flavonoids. A significant difference was observed between varieties and times (*p*-value < 0.05). Total phenols increased with a prolonged germination time, except in QB, whose increase occurred after 48 h. Flavonoids fluctuated in QB and QR, but there was a substantial increase during germination in QN. The different biosynthetic routes (Shikimate and phenylpropanoids) and metabolite synthesis rates could explain the differences between varieties [51,52,60,61,62,63]. This occurs as a protective response when seeds break dormancy during germination [64,65,66,67,68].

### 3.7. Antioxidant Capacity

The antioxidant activity capacity equivalent to Trolox of the quinoa varieties is shown in Figure 4; the values ranged between 11.33 µmol TE/g to 22.62 µmol TE/g for white quinoa, between 12.59 µmol TE/g to 25.28 µmol TE/g for black quinoa, and between 12.39 µmol TE/g to 16.18 µmol TE/g. Furthermore, it was found that most of the samples present significant differences (*p*-value < 0.05), with the antioxidant activity being higher during 72 h, especially for BQ and WQ. The polyphenolic compounds responsible for antioxidant capacity are secondary metabolites present in plants, which are formed during their development and under stress conditions; these include simple phenols, phenolic acids, coumarins, flavonoids, stilbenes, hydrolyzable and condensed tannins, lignans, and lignins [69,70]. Furthermore, these polyphenols could be altered during germination, increasing their content and antioxidant capacity [71].

### 3.8. Correlation of Physicochemical Parameters, Bioactive Compounds, and Antioxidant Activity

The correlation analysis (Figure 5) showed a significant negative association between protein content and lightness values (r = −0.91); this inverse relationship agrees with previous studies that report lower lightness in flours with higher protein content [72], the increase in proteins during germination increases the opacity of the flours by providing more excellent light dispersion, thus reducing lightness values [73]. The chrome *a** (red/green) was positively correlated with proteins (r = 0.76) and negatively with carbohydrates (r = −0.84). This is consistent with the previous point, indicating that the higher the protein content, the higher the reddish tone. The water activity and moisture are strongly correlated (r = 0.83), which makes sense since both parameters are related to the aqueous content [40]. The fat content negatively correlates with the ashes (r = −0.69). This may indicate a dilution effect; the higher the fat content, the lower the ash content. There is a positive correlation between total flavonoids and antioxidant capacity (r = 0.61); flavonoids are a type of phenolic compound with particular chemical structures that confer a high antioxidant capacity [69,74]. The positive correlation indicates that flavonoids are the primary metabolites responsible for the antioxidant capacity in these samples of quinoa. Likewise, the high correlation between proteins and phenolic compounds (0.82) could be seen, associated with stress during germination that increases the levels in the germinated grains.

## 4. Conclusions

Germination significantly increased the content of protein, dietary fiber, mineral micronutrients, total phenolics, flavonoids, and antioxidant capacity in white, red, and black quinoa varieties. The most excellent bioactive compounds and antioxidant capacity increases were achieved after 72 h of germination. Flavonoids showed the most significant correlation with antioxidant capacity, suggesting they are mainly responsible.

Carrying out these studies would provide solid support for the potential use of germinated quinoa flours as a functional ingredient in developing new foods or dietary supplements. Germination is an effective strategy to improve quinoa’s nutritional and functional profile. Therefore, further research in this area is warranted to determine the feasibility of applying this process at an industrial level and harnessing its benefits in developing healthier food products.

In conclusion, the results of this study demonstrate that germination substantially improves quinoa’s nutritional composition and antioxidant capacity. Further studies on bioactive compounds’ stability, bioavailability, and bioaccessibility are required to support the potential use of germinated quinoa flours as functional ingredients in developing foods or dietary supplements. Research in this area would promote using quinoa germination’s benefits at an industrial level.

## Figures and Tables

**Figure 1 foods-13-00417-f001:**
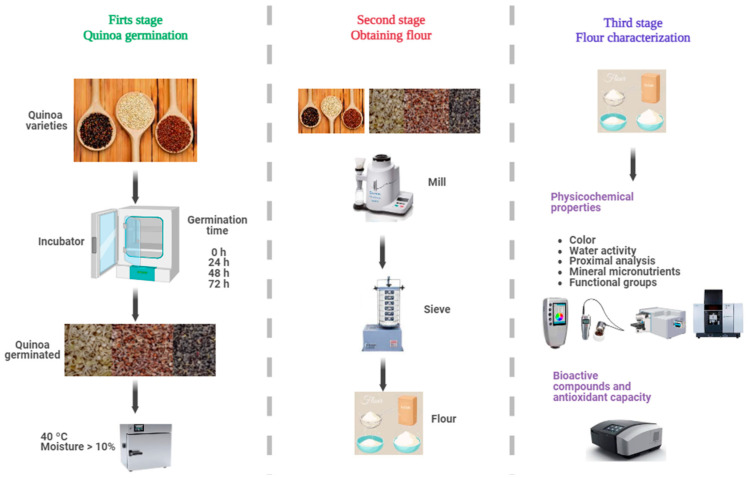
Experimental flow diagram.

**Figure 2 foods-13-00417-f002:**
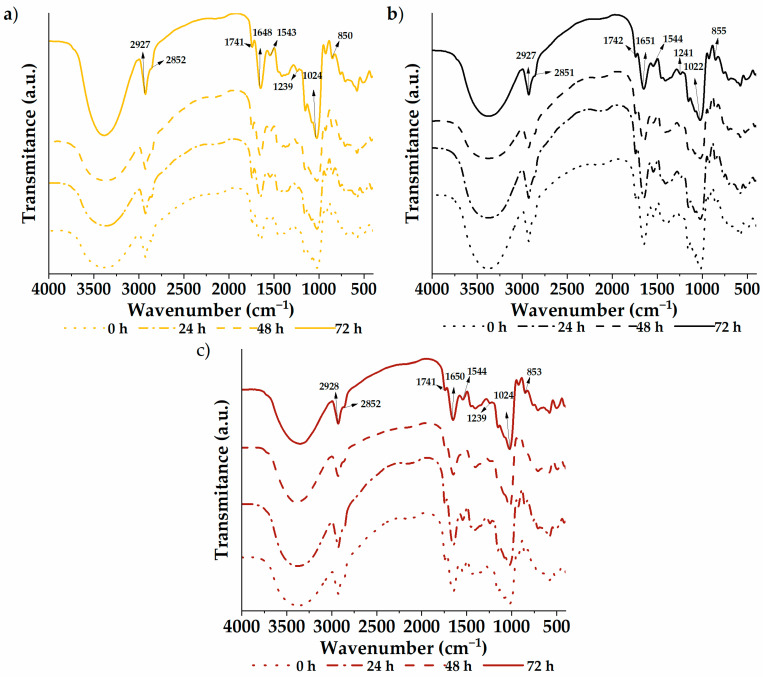
Functional groups: (**a**) white quinoa, (**b**) black quinoa, and (**c**) red quinoa.

**Figure 3 foods-13-00417-f003:**
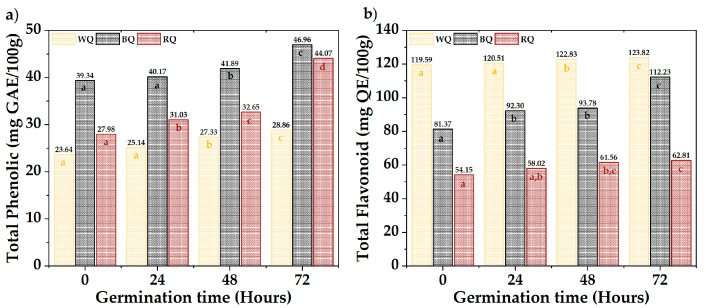
Bioactive compounds, (**a**) total phenolic, and (**b**) total flavonoids. Equal letters mean that there is no significant difference, evaluated through the Tukey test, with α = 5%.

**Figure 4 foods-13-00417-f004:**
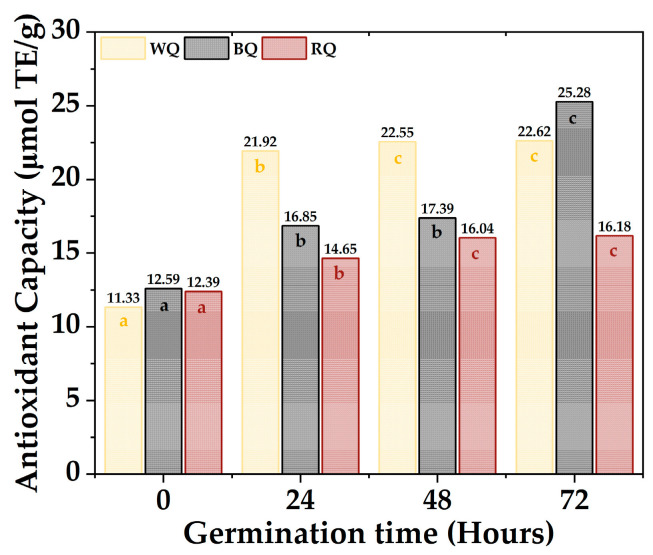
Antioxidant capacity quinoa varieties. Equal letters mean that there is no significant difference, evaluated through the Tukey test, with α = 5%.

**Figure 5 foods-13-00417-f005:**
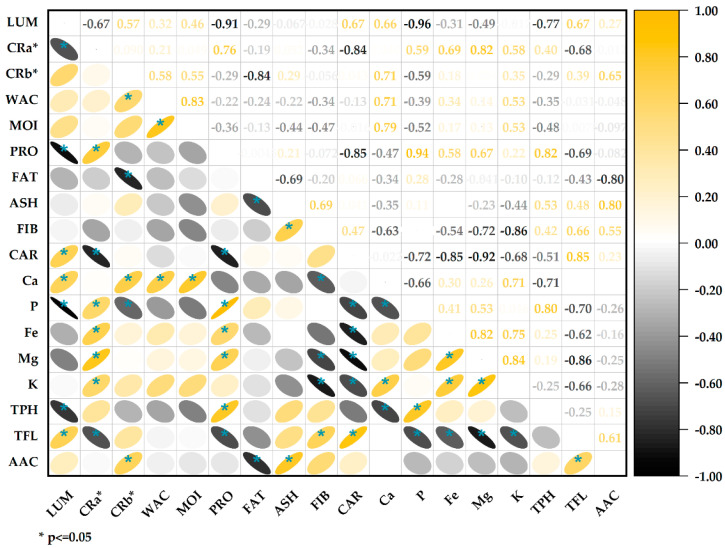
Correlation of physicochemical parameters, bioactive compounds, and antioxidant capacity.

**Table 1 foods-13-00417-t001:** Lightness, chroma *a** and *b**, color variation, and whiteness index.

Variety	Germination Time	Lightness			Chroma *a**			Chroma *b**			Color	ΔE			WI		
		$\bar{x}$	±	s	$\bar{x}$	±	s	$\bar{x}$	±	s		$\bar{x}$	±	s	$\bar{x}$	±	s
WQ	0 h	87.47	±	0.01 ^a^	0.22	±	0.01 ^a^	13.9	±	0.01 ^a^		0.00	±	0.00 ^a^	81.30	±	0.01 ^a^
	24 h	87.57	±	0.00 ^b^	0.58	±	0.01 ^b^	17.7	±	0.02 ^b^		3.83	±	0.03 ^b^	78.37	±	0.01 ^b^
	48 h	87.11	±	0.05 ^c^	0.89	±	0.01 ^c^	19.1	±	0.04 ^c^		5.26	±	0.03 ^c^	76.95	±	0.00 ^c^
	72 h	85.75	±	0.05 ^d^	1.70	±	0.00 ^d^	20.1	±	0.00 ^d^		6.61	±	0.03 ^d^	75.31	±	0.03 ^d^
RQ	0 h	76.01	±	0.02 ^a^	3.24	±	0.01 ^a^	12.1	±	0.01 ^a^		0.00	±	0.00 ^a^	72.95	±	0.02 ^a^
	24 h	79.14	±	0.45 ^b^	2.39	±	0.08 ^b^	13.5	±	0.03 ^b^		3.55	±	0.41 ^b^	75.03	±	0.39 ^b^
	48 h	78.22	±	0.03 ^c^	2.54	±	0.01 ^c^	17.2	±	0.02 ^c^		5.62	±	0.03 ^c^	72.14	±	0.01 ^c^
	72 h	77.64	±	0.00 ^d^	2.74	±	0.00 ^d^	17.6	±	0.02 ^d^		5.76	±	0.02 ^c^	71.43	±	0.02 ^d^
BQ	0 h	73.73	±	0.21 ^a^	1.47	±	0.01 ^a^	9.14	±	0.01 ^a^		0.00	±	0.00 ^a^	72.15	±	0.20 ^a^
	24 h	78.13	±	0.04 ^b^	0.98	±	0.01 ^b^	9.53	±	0.01 ^b^		4.45	±	0.25 ^b^	76.13	±	0.04 ^b^
	48 h	77.01	±	0.81 ^c^	1.18	±	0.07 ^c^	11.2	±	0.02 ^c^		3.92	±	0.58 ^b^	74.38	±	0.73 ^c^
	72 h	74.56	±	0.14 ^d^	3.00	±	0.01 ^d^	16.6	±	0.01 ^d^		7.65	±	0.02 ^c^	69.47	±	0.12 ^d^
*p*-value		0.00			0.00			0.00				0.92			0.00		

Note: $\bar{x}$ is the mean; *S* is the standard deviation; Δ*E* is color difference; *WI* is whiteness index; WQ is white quinoa; RQ is red quinoa, BQ is black quinoa, equal letters mean that there is no significant difference, evaluated through the Tukey test, with α = 5%. The background colors represent the value of *L*, chroma *a** and chroma *b**.

**Table 2 foods-13-00417-t002:** Water activity quinoa varieties.

Variety	Germination Time	Water Activity		
		$\bar{x}$	±	s
WQ	0 h	0.22	±	0.00 ^a^
	24 h	0.10	±	0.00 ^b^
	48 h	0.24	±	0.00 ^c^
	72 h	0.25	±	0.00 ^d^
RQ	0 h	0.23	±	0.00 ^a^
	24 h	0.11	±	0.00 ^b^
	48 h	0.25	±	0.00 ^c^
	72 h	0.24	±	0.00 ^c^
BQ	0 h	0.11	±	0.00 ^a^
	24 h	0.10	±	0.00 ^b^
	48 h	0.12	±	0.00 ^c^
	72 h	0.12	±	0.00 ^d^
*p*-value		0.00		

Note: $\bar{x}$ is the mean; *S* is the standard deviation; RQ is red quinoa; BQ is black quinoa; equal letters mean that there is no significant difference, evaluated through the Tukey test, with α = 5%.

**Table 3 foods-13-00417-t003:** Proximal analysis.

Variety	Germination Time	Moisture (g/100 g)			Protein * (g/100 g)			Fat * (g/100 g)			Ash * (g/100 g)			Fiber * (g/100 g)			Carbohydrate * (g/100 g)		
		$\bar{x}$	±	s	$\bar{x}$	±	s	$\bar{x}$	±	s	$\bar{x}$	±	s	$\bar{x}$	±	s	$\bar{x}$	±	s
WQ	0 h	5.24	±	0.02 ^a^	9.01	±	0.02 ^a^	6.60	±	0.03 ^a^	2.24	±	0.02 ^a^	3.15	±	0.01 ^a^	73.76	±	0.04 ^a^
	24 h	4.72	±	0.04 ^b^	9.74	±	0.01 ^b^	6.48	±	0.01 ^b^	2.33	±	0.01 ^b^	3.20	±	0.01 ^b^	73.53	±	0.05 ^b^
	48 h	5.14	±	0.01 ^c^	9.77	±	0.01 ^c^	6.83	±	0.02 ^b,c^	2.30	±	0.02 ^b^	3.36	±	0.02 ^c^	72.60	±	0.02 ^a^
	72 h	5.37	±	0.02 ^d^	9.97	±	0.02 ^d^	6.44	±	0.02 ^c^	2.40	±	0.02 ^c^	3.80	±	0.02 ^d^	72.02	±	0.06 ^c^
RQ	0 h	5.32	±	0.04 ^a^	12.25	±	0.09 ^a^	6.72	±	0.03 ^a^	2.20	±	0.03 ^a^	2.70	±	0.02 ^a^	70.82	±	0.13 ^a^
	24 h	3.18	±	0.05 ^b^	12.63	±	0.06 ^b^	6.58	±	0.05 ^a^	2.26	±	0.05 ^a^	2.82	±	0.05 ^b^	72.54	±	0.01 ^b^
	48 h	5.24	±	0.06 ^a^	13.35	±	0.07 ^c^	6.66	±	0.20 ^a^	2.28	±	0.03 ^a^	2.88	±	0.01 ^b,c^	69.60	±	0.05 ^c^
	72 h	6.02	±	0.05 ^c^	13.64	±	0.14 ^d^	6.45	±	0.13 ^a^	2.32	±	0.05 ^b^	2.95	±	0.02 ^c^	68.62	±	0.20 ^d^
BQ	0 h	3.63	±	0.04 ^a^	12.41	±	0.05 ^a^	6.80	±	0.04 ^a^	2.20	±	0.01 ^a^	3.60	±	0.05 ^a^	71.36	±	0.16 ^a^
	24 h	3.82	±	0.11 ^b^	12.46	±	0.07 ^b^	6.71	±	0.03 ^a^	2.36	±	0.06 ^b^	3.65	±	0.01 ^a^	71.01	±	0.09 ^b^
	48 h	2.38	±	0.04 ^c^	12.51	±	0.10 ^a^	6.60	±	0.03 ^b^	2.48	±	0.03 ^b,c^	3.70	±	0.07 ^a^	72.34	±	0.10 ^c^
	72 h	3.06	±	0.08 ^d^	14.02	±	0.01 ^c^	6.29	±	0.04 ^c^	2.58	±	0.03 ^c^	3.82	±	0.02 ^b^	70.23	±	0.13 ^d^
*p*-value		0.00			0.00			0.00			0.00			0.00			0.00		

Note: $\bar{x}$ is the mean; *S* is the standard deviation; WQ is white quinoa; RQ is red quinoa; BQ is black quinoa; (*) dry weight; equal letters mean that there is no significant difference, evaluated through the Tukey test, with α = 5%.

**Table 4 foods-13-00417-t004:** Mineral micronutrients.

Variety	Germination Time	Calcium (mg/100 g)			Phosphorus (mg/100 g)			Iron (mg/100 g)			Magnesium (mg/100 g)			Potassium (mg/100 g)		
		$\bar{x}$	±	s	$\bar{x}$	±	s	$\bar{x}$	±	s	$\bar{x}$	±	s	$\bar{x}$	±	s
WQ	0 h	92.20	±	0.02 ^a,b^	170.40	±	0.37 ^a,d^	4.80	±	0.10 ^a^	120.50	±	1.50 ^a^	432.00	±	1.00 ^a^
	24 h	91.70	±	0.04 ^a^	177.50	±	0.40 ^b,c^	4.20	±	0.05 ^b^	142.00	±	1.01 ^b^	437.14	±	0.34 ^b^
	48 h	92.80	±	0.72 ^b,c^	176.40	±	0.74 ^b^	4.55	±	0.05 ^c^	128.30	±	0.30 ^c^	441.60	±	0.30 ^c^
	72 h	93.50	±	0.26 ^c^	178.30	±	0.75 ^c,d^	4.65	±	0.09 ^a,c^	131.50	±	0.10 ^d^	448.30	±	0.61 ^d^
RQ	0 h	88.50	±	0.10 ^a^	328.20	±	1.06 ^a^	5.10	±	0.17 ^a^	190.40	±	0.03 ^a^	510.50	±	1.50 ^a^
	24 h	86.40	±	0.56 ^a^	334.60	±	1.44 ^b^	5.50	±	0.62 ^a^	196.33	±	0.58 ^b^	517.60	±	1.90 ^b^
	48 h	91.30	±	1.13 ^b^	338.40	±	0.60 ^c^	5.42	±	0.33 ^a^	189.00	±	1.73 ^a,c^	520.60	±	0.95 ^b^
	72 h	92.20	±	1.06 ^b^	347.10	±	1.56 ^d^	5.64	±	0.55 ^a^	193.20	±	0.13 ^d^	524.90	±	1.01 ^c^
BQ	0 h	52.80	±	0.35 ^a^	360.40	±	0.52 ^a^	4.05	±	0.05 ^a^	132.60	±	1.22 ^a^	390.10	±	0.95 ^a^
	24 h	55.60	±	0.53 ^b^	370.17	±	0.76 ^b^	4.80	±	0.20 ^b^	140.10	±	0.17 ^b^	398.40	±	0.40 ^b^
	48 h	58.50	±	0.10 ^c^	366.00	±	1.06 ^c^	4.72	±	0.17 ^b^	142.20	±	0.03 ^b^	406.70	±	1.50 ^c^
	72 h	60.20	±	0.35 ^c^	368.80	±	1.31 ^d,b^	5.11	±	0.10 ^c^	156.00	±	1.11 ^c^	402.50	±	1.50 ^d^
*p*-value		0.00			0.00			0.00			0.00			0.00		

Note: $\bar{x}$ is the mean; *S* is the standard deviation; WQ is white quinoa; RQ is red quinoa, BQ is black quinoa, equal letters mean that there is no significant difference, evaluated through the Tukey test, with α = 5%.

## Data Availability

Data is contained within the article.

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
