# Peer review of "Effect of Germination on the Physicochemical Properties, Functional Groups, Content of Bioactive Compounds, and Antioxidant Capacity of Different Varieties of Quinoa (Chenopodium quinoa Willd.) Grown in the High Andean Zone of Peru"

_foods, 2024, doi:10.3390/foods13030417_

Round 1

Reviewer 1 Report

Comments and Suggestions for Authors

Manuscript ID: foods-2806848

Title: Effect of germination on the physicochemical properties, functional groups, content of bioactive compounds, and antioxidant capacity of different varieties of quinoa (Chenopodium quinoa Willd.) grown in the high Andean zone of Perù

The paper presents interesting information on the composition of quinoa sprouts but, in my opinion, the submitted version should be revised.

Material and methods

Lines 104-107. Authors stated that Carbohydrates were determined by difference but I find that the sum of all parameters showed in Table 3 is more than 100% for all samples.

Please, add model and brand of all used instruments (i.e: digital colorimeter; water activity meter; atomic absorption spectrometer etc.)

Results and discussion

Lines 168-139. Authors stated that the recorded changes are attributable not only to germination but also to compounds produced by Maillard reaction. Based on this statement, it can be argued that the adopted drying procedure produce an undetermined bias of the measurements. Why authors do not lyophilise the sprouts to avoid this problem? Otherwise, they should attempt the quantification of color changes attributable to Maillard reaction. Moreover, accumulation of carotenoids should be supported by their quantification. Similarly, the significant correlation between luminosity and protein content (line 273) is related by authors to Maillard reaction not to seed germination.

Lines 153-158 The variation of ΔE recorded for BQ is different as compared to the other two samples. Similarly, WI of RQ and BQ shown different trends respect to WQ. Could authors explain these results?

Line 176 Are data of Table 3 expressed on dry matter base? This important information should be added.

Lines 209-212. I do not agree with this explanation. Hydrolysis of macro molecules cannot affect the amount of minerals being they quantified after complete incineration of samples. The hydrolysis cannot produce the increase of the number of atoms of minerals in the tissues unless they are added during germination trough nutritive solutions Similarly, antinutritional compounds such phytic acid affects the bioavailability of some minerals in sprouts but the capability of chelation is destroyed with tissues incineration.

Line 229-230. The sentence should be revised. The increase of TFC over the germination time is not sufficiently supported by data in Fig. 2. I do not see significant differences for WQ and BQ at different time (i.e. WQ at 0 and 24H, BQ at 24 and 48h).

Minor remarks

Line 27 - 24h is reported two times

Line 153 Color data are shown in Table 1 not 2

Lines 173-174 and 203-204. The notes of the Table 2, 3 and 4 should be revised. They include the parameters of the Table 1 and not those really showed.

Author Response

Dear reviewer, attached is the observation correction report.

Reviewer 2 Report

Comments and Suggestions for Authors

Dear Authors,

The manuscript "Effect of germination on the physicochemical properties, functional groups, content of bioactive compounds, and antioxidant capacity of different varieties of quinoa (Chenopodium quinoa Willd.) grown in the high Andean zone of Peru" - foods-2806848-peer-review-v1, describes the effect of germination on the physicochemical properties, functional groups, content of bioactive compounds, and antioxidant capacity of quinoa. This is an interesting topic, especially as germination is an effective strategy to improve the nutritional and functional quality of grains such as quinoa. Thus, the authors examined the effect of germination at 0, 24, 48, and 72 h on the nutritional and functional quality of three varieties of quinoa: white, red and black, in total 12 experimental variants.

The novelty of the manuscript is clear, and the topic is worthy of investigation. However, some minor changes are necessary, according to the following comments:

Abstract:

line 27: please review [...] evaluated the effect of germination (0, 24, 24, 48, and 72 h) [...], appears twice 24 hours.

Introduction - part should be updated with recent references that justify the topic. Please present in more detail the chemical and functional composition of quinoa, specifying the average values for these.

for example: "[...] it contains significant amounts of high biological value proteins, with all the essential amino acids in ideal proportions for human nutrition." ... please, detailed with examples of compounds and values.

Materials and Methods - They should be described with sufficient detail to allow others to replicate and build on published results. Even though they are well-established methods, they can be briefly described and appropriately cited. For example, "Carbohydrates were determined by difference."... it is not clear enough... briefly present and put the calculation formula. Review all these aspects throughout the work.

Figure 1 - Line 79: please use the order in the image as in the description from methods (line 75) and results (tables)

2.2. Germination

Line 80: "The grains were placed in a humid chamber at 25 °C and 95% relative humidity." Please provide more details for the humid chamber, equipment used, type, country, and city. Similarly, for "forced convection oven" and "cyclone mill" - line 87-89; review these aspects for all methods.

Line 126: please correct 2.11 with 2.9

Line 130: please correct 2.12 with 2.10

Results and discussion

Table 1 - 4; P=0.00??? please check and confirm the correctness of the data.

Line 153: Table 1 instead of Table 2... please check.

Figure 2: please use the order in the figure as in the description of methods (line 75) and results (tables): WQ; RQ; BQ. In the way presented now, it creates confusion for readers.

Line 128-129; Some aspects are not clearly presented, being difficult to understand, for example: "[...] while WB (123.82 mg QE /100g) reported high levels of total flavonoids (123.82)".

Figure 3: please correct similarly to the observation from figure 2.

Line 239-240 “[...] the values ranged between 11.33 to 22.62 for white quinoa, between 12.59 to 25.28 for black quinoa, and between 12.38 to 16.18 µmol TE/g.” ... ??? [....], please check.

References: use italics for genus and species of microorganisms.

Also, the bibliography can be improved by adding additional articles that justify the topic presented at the introduction and improve the discussion part.

Author Response

(The authors gave the same response as above.)

Reviewer 3 Report

Comments and Suggestions for Authors

In this study, the impact of germination duration on the color, water activity, proximal composition, mineral micronutrients, bioactive compounds, and antioxidant capacity of white, red, and black quinoa varieties was explored. The findings underscore the potential of prolonged germination in enhancing the nutritional and functional qualities of quinoa, offering insights into its application in diverse food products. The manuscript is generally well-written; however, there are areas that could benefit from improvement to enhance scientific precision and overall clarity.

Abstract: The abstract lacks a clear statement of the research objectives. The authors are encouraged to provide a more explicit emphasis on the aim and novelty of their study in the abstract. Clearly articulating these aspects would enhance the manuscript's clarity and engage readers more effectively.

Introduction: The introduction effectively outlines the context and importance of quinoa germination but lacks explicit emphasis on the novelty of the study. A more explicit statement on what sets this research apart from previous works would enhance the introduction's impact. Additionally, there is an opportunity to clarify the specific gap in knowledge that the study aims to address, providing a more focused foundation for readers to understand the significance of the research.

Materials and Methods: Include more information on the quinoa varieties used in the study. Specify the equipment used for color evaluation, water activity measurement, and proximal analysis. Additionally, elaborate on the procedures for phenolic compounds and antioxidant activity assessment, ensuring transparency and reproducibility.

Results and discussion:

In the "3.1. Color" section, consider enhancing the presentation of color evaluation results (Table 1). To improve clarity, you might want to supplement the table with color-coded visual aids such as line graphs or bar charts.

In the "3.3. Proximal analysis" and "3.4. Mineral micronutrients" sections: deeper discussion is recommended in terms of the observed variances in carbohydrate, protein, and mineral content, considering linking these variations to specific biochemical processes during germination.

Conclusions: it would be beneficial to include insights into future perspectives and potential avenues for further research.

Author Response

(The authors gave the same response as above.)
